# Nest Entrances, Spatial Fidelity, and Foraging Patterns in the Red Ant *Myrmica rubra*: A Field and Theoretical Study

**DOI:** 10.3390/insects11050317

**Published:** 2020-05-21

**Authors:** Marine Lehue, Claire Detrain, Bertrand Collignon

**Affiliations:** Unit of Social Ecology, Université libre de Bruxelles, 1050 Brussels, Belgium; marine.lehue@ulb.ac.be (M.L.); claire.detrain@ulb.ac.be (C.D.)

**Keywords:** collective behaviour, foraging, nest architecture, social insects

## Abstract

The nest architecture of social insects deeply impacts the spatial distribution of nestmates their interactions, information exchanges and collective responses. In particular, the number of nest entrances can influence the interactions taking place beyond the nest boundaries and the emergence of collective structures like foraging trails. Here, we investigated in the field how the number of nest entrances impacted the foraging dynamics of *Myrmica rubra* ant colonies. We located the nest entrances where recruitment occurred towards sugar feeders placed in their surroundings. The nests showed one or multiple entrance(s) aggregated in clusters spaced by at least 15 cm. Foragers from colonies with two clusters of entrances were distributed more homogeneously among the feeders than those of colonies with one cluster. In addition, foragers always returned to the first discovered feeder and demonstrated a high fidelity to their original entrance. Finally, a multi-agent model highlighted that additional entrances and clusters of entrances delayed the mobilisation of workers but favoured the simultaneous exploitation of several sources, which was further enhanced by the spatial fidelity of foragers. Multiple nest entrances seem to be a way for medium-sized colonies to benefit from advantages conferred by polydomy while avoiding associated costs to maintain social cohesion.

## 1. Introduction

Social foraging has been described in many species including fish, birds, mammals and insects (reviewed in [1]). Cooperation during food exploitation usually results in an increase of food intake per individual compared to solitary strategies. This can be achieved by the sharing of information about resources [2,3,4], the sharing of food [5,6], and a higher efficiency to catch preys or retrieve food sources [7,8]. Collective foraging has been extensively studied in social insects as it raises challenging questions about the mechanisms underlying the coordination of hundreds of individuals in the absence of any task managers, or centralized decision-making [9,10]. In most ant species and in honeybees, collective foraging is based on the sharing of information through recruitment of naive individuals by informed ones [11,12] and on the transfer of food from foragers to workers, larvae, or queens that stay inside the nest. [13,14].

Collective foraging in social insects usually relies on self-organized processes in which individual behaviours are regulated through positive and negative feedbacks that amplify or dampen the emergent group behaviours [15]. The emergence of these collective behaviours requires a critical number of interactions among individuals to take place [10,16]. Reaching a threshold of interacting individuals implies that numerous trade-offs have to be solved or optimized by the colony. These include (i) how to divide the workforce between scouts searching for food and recruits waiting in the nest for information [17], (ii) how long should recruits wait for information before leaving the nest to find food on their own [18], (iii) how to maximize food discovery and the area explored around the nest in a given time/energy budget [19], (iv) how should information about different resources brought by the individuals be processed to best fit the colony needs [15], and (v) whether the foragers should aggregate in a single place or split in different nest subunits [20,21].

These two last questions are closely linked to the number and location of nest entrances where information and food can be shared between nestmates and foragers. In central place foragers, such as ants, workers aggregate within the entrance chamber where retrieved food items and information are exchanged through trophallaxis, pheromones or antennal contacts [22,23]. Moreover, the density of ants at the entrance will determine the rate of encounters with successful foragers and hence the rate at which potential recruits will leave the nest [24,25,26,27]. This makes the entrance chamber a key location that potentially shapes the colony foraging response. In the meantime, entrances also appear as highly flexible structures, of which the location and number vary depending both on changes in the underground spatial organization of workers [28] and in the distribution and quality of food sources present in the outside environment [29,30].

Regarding the functional value of a nest with multiple entrances, a partitioning of foragers between entrances may reduce the average distance that foragers have to walk outside the nest before reaching a food item. This may also contribute to enlarge the area that foragers cover, thereby decreasing the energy/time costs of food searching and food collection [30,31]. Finally, multiple entrances provide shelters while foraging and lower risks of being predated or threatened by competing ant colonies [29,32,33].

As suggested by theoretical [20] and experimental [21] studies, the energetic and nutritional gains that one species may expect from multiple access points during foraging are highly variable and closely dependent on the distribution and type of exploited food sources. Colonies possessing a single entrance chamber would increase their foraging success when they collectively exploit clumped and patchy food sources, because information sharing and mobilization is enhanced when a dense population is available for recruitment at a single central location [20]. Conversely, in polydomous colonies possessing multiple sub-nests, the foraging success increases when food is scattered, due to their higher ability to quickly find food sources during the phase of exploration [20,34]. One may wonder whether the multiplicity of nest entrances would influence the dynamics of foragers mobilization and the number of discovered and exploited resources by the colony, in a way analogous to polydomy [20], albeit at a smaller spatial scale.

To test this hypothesis under field conditions, we studied the foraging behaviour of *Myrmica rubra* colonies, as their nests regularly show multiple entrances and their omnivorous diet includes food types, such as prey or aphid honeydew, that vary in number and spatio-temporal availability. We investigated the impact of the number and position of the nest entrances on the recruitment and spatial distribution of workers among multiple food sources. Furthermore, the level of spatial fidelity of foragers to the first food source encountered, as well as the first entrance exited, was assessed since it was likely to shape both the foraging dynamics and the spatial distribution of workers among resources. Based on these field data, we developed a model to study how the number and clustering of nest entrances, as well as the spatial fidelity of foragers, may influence the foraging patterns of ant colonies. This model allowed us to test the hypothesis deduced from our experiments in a controlled environment and to extend our conclusions to other nest configurations and food types.

## 2. Materials and Methods

We studied five nests located on the university campus (Brussels, Belgium). For each nest, we delineated a 4 m^2^ quadrat around the first identified entrances and separated this area into a grid composed of 16 squares of 0.25 m^2^ each (Appendix A). In this study, an “entrance” will refer to any functional hole out of which ants exit their nest to forage in the external environment, regardless of its proximity to another hole. A “cluster” will refer to several entrances being aggregated at smaller distances than 15 cm (see results below supporting this distance threshold value). For some colonies, the number and/or the position of nest entrances changed during the study period, so the entrances were not always centred in the grid. Since *M. rubra* is well-known to be aggressive against ant foreigners, we performed dyadic encounters to identify the different colonies while drawing the map of nest entrances. Recordings and observations of the ants started at 12:00 solar time. We performed two experiments. The first one studied the collective exploitation of multiple sugar food sources and the second one assessed the fidelity of the foragers to specific entrances and food sources.

### 2.1. Collective Exploitation of Food Sources

We monitored the foraging activity of the colonies for a total of ten observations. As some colonies were not discovered from the beginning of our study, we did not perform the same number of observations for all colonies (for a summary of the number of observations for each colony, see Appendix A). For each colony that we observed more than once, seven days elapsed between each observation. We placed eight food sources at regular intervals, 50 cm apart from each other, on 8 intersections of the grid, forming a square within the observed area (Appendix A). Each food source consisted in a cotton pad saturated with 0.3 M sucrose solution. Such a sucrose concentration is of the same order of magnitude as the sugar concentration found in aphid honeydew [35] and stands above the food acceptance threshold of many ants including *M. rubra* species [36,37]. Once the food sources were deposited, we recorded the foraging activity for 120 minutes as follows: (1) we identified the position of active entrances by following foragers, (2) we counted the number of ants present at each source every 15 minutes, (3) we computed the distance between each food source and each entrance, (4) we measured the time foragers spent walking between food sources and entrances. For the latter measure, we focused our observations on sources where at least 10 ants could be tracked from their departure from the resource until their arrival at an entrance. If the 10 tracked ants went to the same entrance, we stopped the recordings for the corresponding source. If the returning ants distributed themselves among several entrances, we tracked supplementary ants until obtaining at least 5 tracked ants per entrance. We tracked a total of 279 ants (70 ants for colony 1, 69 ants for colony 2, 32 ants for colony 3, 60 ants for colony, and 48 ants for colony 5). As this monitoring was time consuming, we were not able to track simultaneously foragers of different colonies during the same day. Therefore, we measured the duration of the foragers’ journeys in two out of three observations for colony 1, two out of three observations for colony 2, one out of one observation for colony 3, one out of two observations for colony 4, and one out of one observation for colony 5.

Finally, we quantified the distribution of ants between food sources over time using the Pielou’s evenness index:
$$H=\frac{-\sum_{i=1}^{S}p_{i}\ln p_{i}}{\ln S}=\frac{-\sum_{i=1}^{8}p_{i}\ln p_{i}}{\ln8}=-\overset{8}{\underset{i=1}{\sum }}p_{i}\log_{8}p_{i}$$
with *S* being the number of food sources (eight in our experiments), *p_i_* being the proportion of ants at food source *i*, calculated on the total number of foragers observed at all food sources. A Pielou’s evenness index at 0 indicated that all ants were focused on a single source, and an index of 1 indicated that they were evenly distributed between all the sources. The impact of the number of nest entrance/clusters on the number/partitioning of foragers among feeders was analysed statistically by considering all the 10 observations as independent data. Indeed, when successive observations were made on the same colony, they were separated by at least 7 days and the network of active entrances was never the same (except for one colony that used twice the same couple of entrances).

### 2.2. Ant Fidelity to Nest Entrances and to Food Sources

To assess the spatial fidelity of foragers, we performed experiments on colonies consistently having multiple active entrances. Thus, this experiment was carried out twice on colonies 2 and 4 for a total of 4 observations (three of the colonies had only one active entrance at that time and were thus excluded). We placed two food sources at 40 cm on both sides, perpendicularly to the average axis joining the nest entrances. During the first hour, we individually tracked foragers from their exit of one entrance until their arrival to one of the two food sources. There, each tracked ant was marked with a drop of paint on its abdomen while eating at the food source. We used one colour for each entrance/source combination that was defined both by the entrance exited and by the food source reached by the tracked ant. We stopped marking ants for a particular entrance/source combination as soon as 10 to 15 ants had been marked with the corresponding colour. In the second hour of the experiment, we counted, every 10 minutes, the number and colour of marked ants observed at each food source. Simultaneously, we continuously scanned entrances to measure the number of marked ants entering the nest entrances. In total, we paint-marked 132 ants for a total of 14 different entrance/food combinations (see Appendix A for more details).

### 2.3. Model

We developed a multi-agent model to simulate the foraging activity of an ant nest with multiple entrances. We investigated how the partitioning of the ant population into sub-units connected to different entrances impacted the colony efficiency to forage on different types of food sources. To this end, we simulated a colony of 90 agents hosted in nests differing by the number and arrangement of entrances. This total population was equally distributed in subgroups among the different entrances out of which they could access the external environment and/or be recruited by successful foragers. In the field, the nest entrances were usually distributed in one or two clusters that were spaced by at least 15 cm, each cluster containing one to four entrances. Based on this observation, we simulated different configurations of entrances to decipher how the number of entrances and/or of clusters respectively influenced the foraging efficiency towards different resources. We organised entrances into (i) one cluster of one, three, or six entrances, (ii) two clusters of one, two, or three entrances (iii) three clusters of one or two entrances. For each configuration, we simulated two scenarios: (i) the presence of eight unlimited resources (∞ food units) as in our field experiments; (ii) the presence of a unique and limited resource (600 food units). The food sources were distributed randomly at the beginning of each simulation. In addition, for each scenario, we studied the impact of forager fidelity to the first food source that they exploited. The impact of the entrance configuration on foraging efficiency was assessed by the number of ants present at each resource over time (scenario with eight unlimited food sources), or the time required to collect the entire food source (scenario with a unique and limited food source). The model was run 50 times for each permutation of the number of entrances and clusters and each simulation lasted 7200 timesteps. In the first scenario, we also computed the Pielou’s evenness index to characterize the distribution of foragers among multiple resources. The results were fitted using generalised mixed models with the number of clusters and the number of entrances as fixed effects. For a detailed description of the model, see Appendix A.

## 3. Results

### 3.1. Field Experiments

#### 3.1.1. Collective Exploitation of Food Sources

For each observation, we identified the entrances used by foragers during the two-hour period of food exploitation (Appendix A). The number of active entrances per colony ranged from one to six, with an average of 2.9 ± 1.5 entrances per colony (mean ± SD, n = 10). A 10-minute monitoring of the flows of workers during the four following days revealed that 86% of entrances were active every day (Appendix A). The use of nest entrances was quite variable over time. Among the three colonies that we followed for several weeks in a row, only one colony (colony 4) kept exactly the same configuration of active entrances during two consecutive weeks. The two other colonies (1 and 2) used different entrances from one week to the other (e.g., between week 1 and 2 of colony 1), with on average 3.25 entrances being newly used or being abandoned each week (Appendix A). For each observation, we computed the distance between each couple of entrances. The distance lengths followed a bimodal distribution with a first local maximum between 5 and 7.5 cm and a second one around 30 cm (Figure 1). This bimodal distribution of distances indicated that some entrances were aggregated into clusters, each cluster being separated by several decimetres. When we sorted the entrances into clusters with a threshold distance of 15 cm, we found that the entrances were always aggregated into one or two clusters for each colony (Appendix A) and that recruitment occurred at entrances located in 1.6 ± 0.49 (mean ± SD; n = 10) different clusters.

With respect to the dynamics of food exploitation, the total number of ants at all feeders increased slowly during the first 15 minutes of observation. Then, the number of feeding ants rapidly escalated until a plateau value of 60 ants on average during the second hour of foraging (Figure 2A). This non-linear increase of the number of foragers typically results from the recruitment process that underlies cooperative foraging in ants. Concerning the distribution of ants between food sources, the Pielou’s evenness index increased sharply during the first 30 min of foraging, indicating that the ants progressively discovered and exploited several sources simultaneously (Figure 2B). The index kept on increasing until 60 min and stabilized during the second hour of observation at approximately *H* = 0.6. This value indicated that, instead of focusing their activity on a single resource, ants tended to extend their foraging range, to increase the number of exploited sources, and to distribute themselves among several resources over the course of the recruitment.

We looked whether the number of entrances or clusters of entrances influenced (i) the cumulated number of foragers observed at all feeders or (ii) the partitioning of foragers among feeders. The number of active entrances was not correlated to the cumulated number of foragers at feeders (Spearman correlation, *r* = 0.07, *p* = 0.84, Appendix A), nor to their distribution among the different food sources (Spearman correlation, *r* = 0.43, *p* = 0.21, Appendix A). Likewise, the cumulated number of ants at feeders did not differ between colonies showing one or two clusters of entrances (Mann–Whitney, *U* = 7, *p* = 0.17, Appendix A). However, we found that foragers distributed themselves significantly more homogeneously among feeders when entrances were grouped in two clusters rather than in a single cluster (Mann–Whitney, *U* = 2, *p* = 0.02, Figure 2C).

Finally, we ranked each food source according to (i) the distance to its closest entrance and (ii) the cumulated number of foragers observed at this food source. By relating these rankings, we found that ants exploited significantly more intensively the food sources that were closer to the entrance they exited (Appendix A, Kendall’s t = 0.394, *p* > 0.001). We also computed the duration of the forager’s journeys between the feeders and the entrances. The ranking of one trip observation was not related to its duration (Kendal test: tau = −0.072, *p* = 0.075), meaning that the walking speed was not biased over time due to a possible short-term memory of experienced ants (if any unmarked ant was by chance followed twice in the same session). The time to reach a feeder was linearly correlated to its spatial proximity (linear regression, slope = 1.68, *r* = 0.77, *p* < 0.001, Appendix A) and the speed of ants (0.43 ± 0.15 cm.s^−1^, mean ± SD) was not influenced by the travelled distance.

### 3.1.2. Ant Fidelity to Nest Entrances and Food Sources

We studied ant fidelity to the first entrance they exited and to the first food source they visited. Among the 132 foragers that were marked after their first trip with a colour specific to each entrance/food source combination, we re-observed only 116 marked ants at feeders during our periodic scans, meaning that not all foragers exited the nest a second time, or that some were cleaned from their marking while in the nest. All foragers were re-observed at the food source where they were initially marked, indicating a remarkably high fidelity of 100% to the first feeder they visited (Figure 3). In the meantime, we continuously monitored the number of times that ants went back to their initial entrance. We observed a total of 254 returns of marked ants meaning that, on average, marked ants were seen 1.93 times returning to one of the 14 nest entrances (254 observations at nest entrance out of 132 marked ants). Only three out of 254 ants came back to entrances that were located outside their cluster of departure. Among all the ants that returned to their cluster of departure, 210 out of 251 ants returned to the entrance they first exited (Figure 3). Thus, the ants demonstrated a very performant homing behaviour as well as quite remarkable fidelity to a specific nest entrance.

### 3.2. Model

First, we simulated conditions similar to our field experiments, with eight unlimited resources distributed in the environment. We found that the total number of entrances altered the dynamics of nestmate recruitment. Indeed, the fastest mobilization was observed in colonies showing a single nest entrance (Figure 4A). Additional entrances slowed down the mobilisation of foragers, independently of the number of clusters that they formed. The number of clusters had also an impact on the distribution of the ants among the different food sources. For a given number of nest entrances, increasing the number of clusters from one to three resulted in higher values of evenness index, indicating that ants simultaneously exploited more food sources (Figure 4B).

Coupled together, a large number of entrances and of clusters increased the average final value of evenness index but also reduced the variability of its distribution (Figure 5, GLM, Pielou’s evenness index ~ Entrances × Clusters, effect of the number of entrances: statistic-value = 2.190, *p* = 0.024; effect of the number of clusters: t-value = 4.994, *p* < 0.0001, interaction effect: t-value = −1.636, *p* = 0.10). This indicated that multiple entrances and clusters associated to a high spatial fidelity of foragers make very likely the simultaneous discovery and exploitation of several resources. Furthermore, we ran a second set of simulations in which foragers no longer showed a fidelity to the first discovered food source and oriented themselves similarly to naive ants, i.e., only on the basis of the pheromone present in the surrounding cells. The removal of this fidelity to a feeder always reduced the value of the Pielou’s evenness index for a given configuration of nest entrances (Figure 5).

In the case of multiple resources, simulations showed that an increasing number of entrances and clusters favoured their simultaneous exploitation but slowed down the dynamics of recruitment. Additional simulations showed that the number and configuration of entrances also influenced the foraging efficiency when ants exploited a single and limited food source of 600 units. Since the food source was exhausted at the end of all simulations, the time spent to collect the food was used as a proxy for foraging efficiency. In this case, multiple entrances and clusters reduced the foraging efficiency of the colony by lengthening the time for complete food harvesting (Appendix A, GLM, Collection time ~ Entrances × Clusters, effect of the number of entrances: statistic-value = 4.872, *p* < 0.0001; effect of the number of clusters: t-value = 2.037, *p* < 0.05, interaction: t-value = −1.812, *p* = 0.07).

## 4. Discussion

This study shows that the number of nest entrances and their spatial configuration influenced the collective foraging response of *Myrmica rubra* colonies. In the field, the entrances were aggregated into one or two clusters, with two clusters significantly favouring the simultaneous exploitation of multiple food sources. While the number of entrances did not significantly influence the number of mobilized foragers in the field, theoretical simulations suggest that, for a fixed population size, additional entrances can delay food recruitment and favour the simultaneous exploitation of multiple food sources. In our observations, the ants also tended to exploit the closest resources and showed a high fidelity to the first food source they exploited and to the entrance they exited.

In the field, the number of entrances per nest ranged from one to eight and were never separated by more than 50 cm. This limited distance indicates that the underground nest structure was not spread over a large area. Nest entrances were aggregated into one or two clusters rather than being evenly spaced. Similarly, in the Australian meat ant *Iridomyrmex* spp., the nest entrances of a single colony spread into a dozen of clusters, each of them formed by up to ten entrances [30,38]. Our study also revealed that the foraging activity of *M. rubra* colonies could shift from one cluster of entrances to another in a week time. Few data are available about the duration of nest entrances, that may range between one and three months (e.g. *Crematogaster* sp, *Iridomyrmex* sp, *Meranoplus* sp, *Pheidole* sp, *Rhytidoponera* sp), with an annual turnover of 2–10 entrances [28]. As regards *M. rubra* nest entrances, the heterogeneous distribution of environmental resources or the presence of competitors may lead the colony to disregard some entrances that could ultimately be clogged [39,40]. Such a variable configuration of multiple entrances may help the colony to cover a large foraging area and facilitate the access to different foraging opportunities. In a one-year field survey of the Australian ant community [28], up to 40% of the ground was within 10 cm of a nest entrance, due to their high turnover. Thus, multiple entrances enable ant colonies to respond quickly to environmental changes by redistributing the workforce on specific entrances or by modifying their spatial location according to internal factors (i.e. the structure of the underground galleries) and external factors (food opportunities or competitors).

Regarding the mobilisation of recruits, the total number of foragers on feeders followed a logistic growth until reaching a plateau, which is typical of mass and group-mass recruiting ant species [41,42]. In our observations, the plateau value was not influenced by the total number of nest openings or clusters. Thus, the number of entrances was not limiting or enhancing the number of foragers that were actually recruited by successful scouts. This result contrasts with field observations of other species (*Iridomyrmex* sp.) that showed a relationship between the number of nest entrances and the population size of foragers [43].

In our field experiments, the exit of recruits seemed to be independently activated at each nest entrance. Indeed, a recruiter entering an entrance did not induce the mobilisation of workers at the neighbouring exits, even those belonging to the same cluster. Likewise, lead casts of *Iridomyrmex purpureus* nests showed that each entrance was linked to a specific set of galleries, with a low connectivity between them [38]. While we did not study the internal nest structure, our observations also suggest that superficial chambers were not highly connected to each other, or at least did not favour the propagation of recruitment among nearby exits. Based on our simulations, the dispatching of the recruits in several unconnected entrance chambers would delay the mobilisation of the workforce, even though the same number of foragers were ultimately recruited. This slower dynamics was observed for several food availabilities, i.e. multiple unlimited resources or a single limited one. These results are consistent with previous works on polydomy that highlighted the higher efficiency of a single nest to collect a single resource compared to five smaller nests [20,21].

Regarding the distribution of foragers among available resources, *M. rubra* colonies did not focus their workforce on a single one but simultaneously exploited several sources, in particular those located closer to their nest entrance. Foragers distribution was more homogeneous when the entrances were spatially organized into two clusters. Likewise, our simulations showed that multiple entrances and clusters favoured the even and simultaneous exploitation of several resources. Indeed, two or three clusters, and to a lesser extent several nest entrances, led to higher average values of the Pielou’s evenness index and reduced the variability of these indexes across simulations. From a functional perspective, a scattering of the workforce on several food sources could be seen as a bet-hedging strategy against changing conditions of food availability or food quality. Multiple entrances also provide a safeguard at the collective level against the investment of all available foragers on a single resource, which may jeopardize the discovery and exploitation of other more profitable resources, or impede the ability of the colony to reorient its workforce following a decrease in food profitability [44].

The homogeneous distribution of *M. rubra* foragers among several food sources contrasts with the collective choices reported for other mass recruiting ants observed in the lab whose workers faced with two identical feeders focused their activity on a single food source [41]. However, in the latter experiments, ants were hosted in one-entrance nests and foraged on a Y shaped bridge that forced trails to converge at a choice point. These converging trails allowed ants to compare their respective pheromone concentration and facilitated the collective selection of one feeder. Conversely, the multiplicity of entrances of natural *M. rubra* nest decreased the probability that scouts would lay trails that converged either on the foraging area or at the same entrance. This reduced the ability for potential recruits to locally compare different recruiting stimuli and thus hampered the collective choice of a single food source (Lehue & Detrain, in prep). Instead, multiple entrances where information sharing and recruitment take place, maintain a diversity of foraging paths followed by recruits and hence a homogeneous repartition of feeding ants among several food sources.

Our field study also revealed a strong fidelity of ants both to a specific nest entrance and to the first food source they encountered. *M. rubra* ants thus possessed high abilities to memorize the spatial location of their nest entrances resulting in an efficient homing behaviour (82% of returns to their initial entrance). Almost all ants that did not come back to their initial entrance did it in a nearby entrance (at less than 15 cm distance). Such a spatial fidelity was reported in other ant species (e.g., *Lasius neoniger*), whereby foragers showed strong fidelity to a specific nest entrance and to a particular area of foraging [32]. In addition, ants showed a 100% fidelity to the first food source they exploited, regardless of whether they discovered the feeder on their own or after being recruited. At the level of individual foragers, the fidelity to specific nest-source trajectory may be suboptimal when the target entrance is not the closest one to the discovered food source. However, the optimization of individual paths (i.e., coming back to the closest entrance) would require high cognitive skills of foragers such as a mental map of all nest entrances or the ability to determine the position of all entrances according to the current location of the forager. The possibly longer distances travelled by ants can also be compensated by an improvement of ants’ navigation skills on better-known areas [32,45,46] and an increase of individual success rates over successive foraging trips [47]. 

Simulations showed that spatial fidelity favoured the simultaneous exploitation of multiple resources, making the colony potentially more resilient to changing food availabilities. Ants’ fidelity to a specific food source could nevertheless impede the ability of the colony to reorient its foraging activity towards more profitable or newly discovered resources [48]. Indeed, the flexibility of foraging patterns depends not only on the recruitment type (trail-based VS leader-based recruitment, e.g. in *Tetramorium caespitum* [41,49] or on the physico-chemical properties of the recruiting signal [50], but also on the ability of the foragers to update their private information (i.e. memory) with public cues laid by other workers (i.e., pheromone trails) [51,52,53,54]. Further research is required to investigate whether the spatial fidelity of *M. rubra* foragers could be tuned down or up as a function of the quality of the resources, with foragers being less eager to give up energetically rewarding resources, stable or larger food sources [55,56,57].

As regards the ecology of *Myrmica rubra* ants, this species is a moderately opportunistic species that feeds on large and patchy resources such as aphid honeydew, as well as on small and scattered prey items [58,59]. Such prey items were retrieved by foragers whose returns into the nest did not triggered the exit of additional workers, thereby preventing a useless mobilisation towards items that can be transported individually [59,60]. In this context, the spatial configuration of nest entrances that we observed in the field may be well adapted to the different foraging strategies that *M. rubra* colonies can concurrently use to exploit available resources. Indeed, an intermediate number of clusters and entrances appears as a trade-off between a rapid mobilization of the workforce towards large food sources and the potential to diversify the exploitation of several scattered food sources. Future studies should also investigate how the spatial configuration of nest entrances can influence foraging efficiency in a dynamic environment with limited-rate resources which deplete and refill, such as honeydew from aphid patches [54].

## 5. Conclusions

There are connections between multiplicity of nest entrances and polydomy in terricolous ant species. In the case of polydomic species, multiple nesting sites make the colony less reliant on the survival of any particular nest, reduce costs of food transport and enables the colony to diversify food sources by foraging over larger areas [31,61]. Our field data on *M. rubra* suggest that this latter advantage could also be conferred by multiple entrances which allocate workers more evenly throughout a colony’s foraging area. The coupling of high spatial fidelity to multiple nest entrances may be seen as a kind of small-scale polydomy. In this context, multiple exits improve the colony ability to explore a wider area and spread risks over several resources, but allow to maintain in a single nest an ant population that remains large enough to generate a collective exploitation of food sources. While additional observations on different ant species should supplement this work, multiple nest entrances appear as a way for small- or medium-sized colonies to benefit from some of the advantages conferred by polydomy while avoiding the associated costs of dispersing resources across multiple nests or maintaining social cohesion through the inter-nest traffic of workers.

## Figures and Tables

**Figure 1 insects-11-00317-f001:**
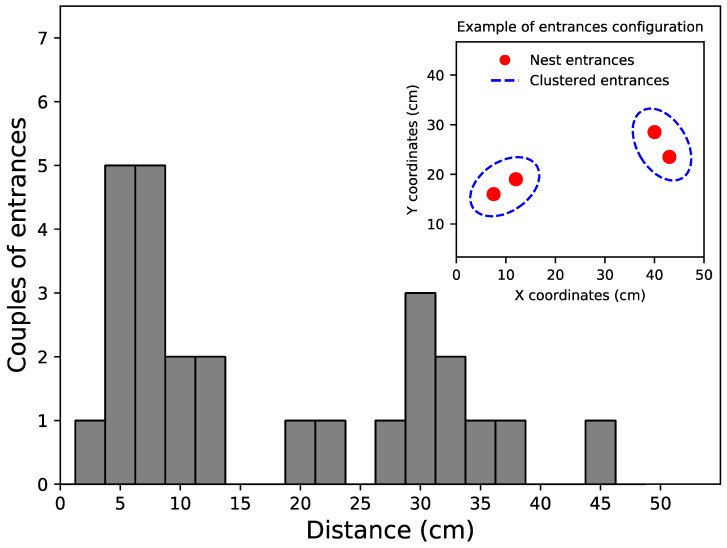
Distribution of the distances measured between all couples of entrances in our field experiments (n = 35) and example of entrances’ configuration observed in the field (inset). The distance values follow a bimodal distribution, indicating that entrances were aggregated into one or two clusters separated by a threshold distance of 15 cm.

**Figure 2 insects-11-00317-f002:**
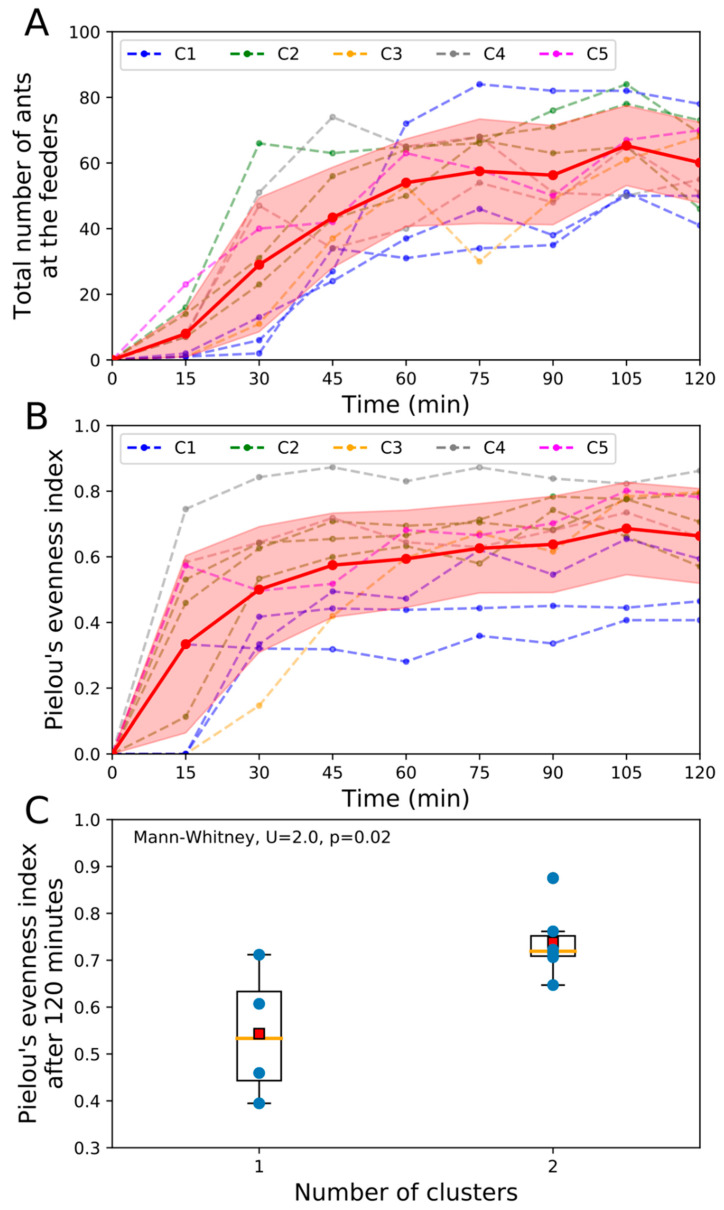
*Exploitation dynamics of food sources*. (**A**) Total number of ants present on the food sources as a function of time. (**B**) Pielou’s evenness index’s values as a function of time. Dashed lines and full circles represent the dynamics of the number of ants at the food sources for each colony (in blue (Colony C1), green (C2), yellow (C3), grey (C4) and pink (C5). Dashed lines of the same colour represent the multiple observations made on the same colony. The mean total numbers of feeding ants that were averaged over all observation sessions are represented in red (circles and line) and standard deviation in red shading. (**C**) Influence of the number of clusters on the Pielou’s evenness Index measured at the end of the experiment. For each boxplot, the red square and the orange line represent the mean and the median of the data distribution, respectively.

**Figure 3 insects-11-00317-f003:**
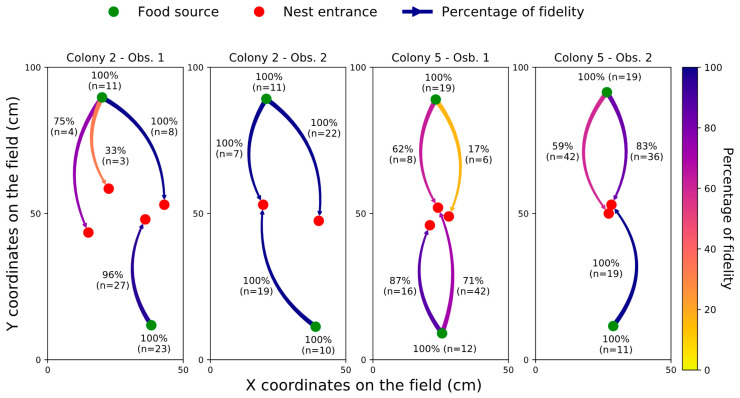
*Fidelity of foragers to the entrance they first exited and to the food source they first exploited*. The label of the arrows indicates the proportion of ants returning to their entrance (red dots) of initial departure and, in brackets, the total number of tracked journeys (n). The label of the food sources (green dots) indicates the proportion of foragers returning at the feeder that they had first visited and, in parenthesis, the total number of marked ants observed at this feeder (n).

**Figure 4 insects-11-00317-f004:**
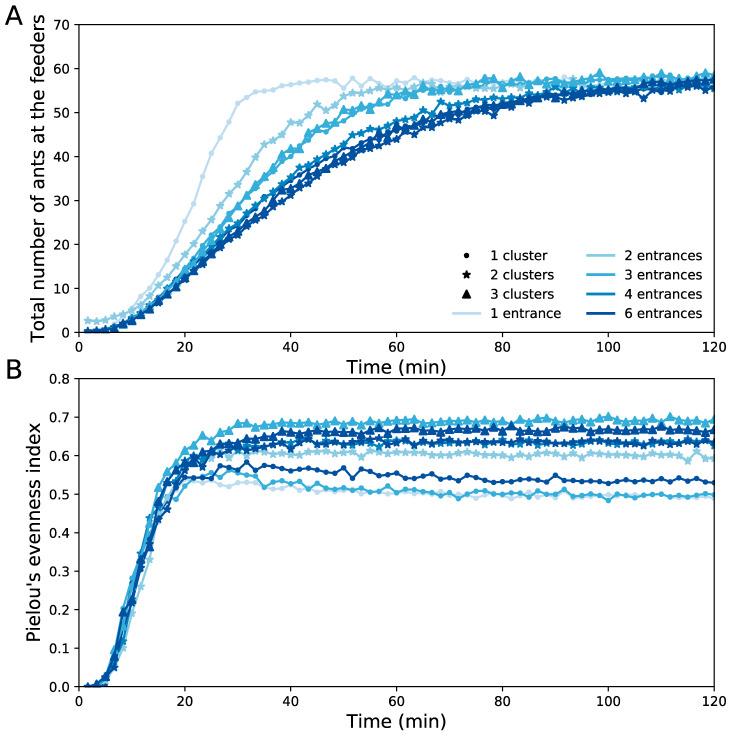
*Theoretical dynamics of the exploitation of eight food sources*. (**A**) Total number of ants present on the food sources as a function of time. (**B**) Pielou’s evenness indices accounting for the distribution of ants among the feeders as a function of time. Symbols represent the total number of clusters (dot = 1, star = 2, triangle = 3) and the gradation of colours represents the total number of entrances (one, two, three, four, or six).

**Figure 5 insects-11-00317-f005:**
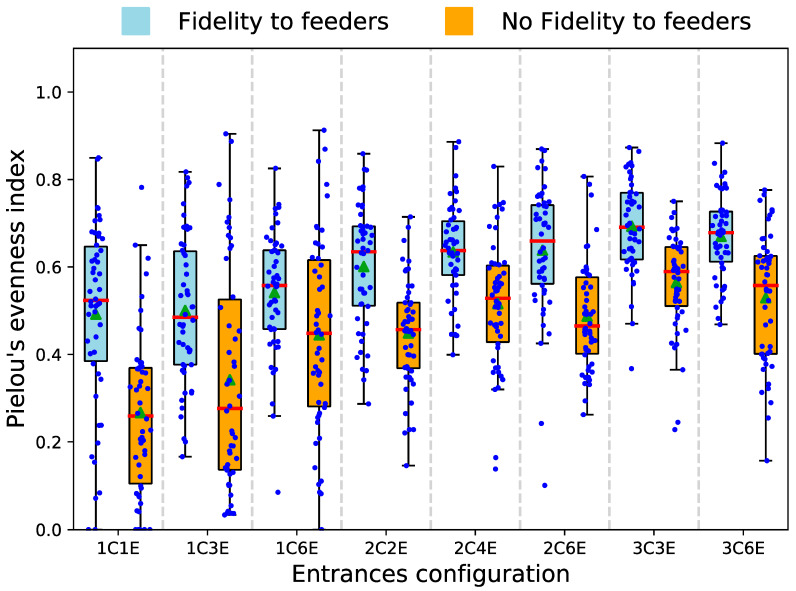
*Influence of the nest entrances configuration* (n = 50 simulations for each configuration) *on the distribution of foragers among eight food sources.* Distribution of the Pielou’s evenness index at the end of the simulations for different entrance configurations. We evaluated the impact of ant fidelity to the first encountered food source by simulating foragers moving with a memory bias (blue boxplot), or without it (orange boxplots). Green triangles indicate the mean, red lines indicate the median and circles indicate outliers.

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
