# Peer review of "Nest Entrances, Spatial Fidelity, and Foraging Patterns in the Red Ant Myrmica rubra: A Field and Theoretical Study"

_insects, 2020, doi:10.3390/insects11050317_

Round 1
Reviewer 1 Report
This is a nice little study on how distribution of nest entrances in an ant colony can affect the exploitation of a food resource (both in the field and in a simulation). The sample size is a bit small, so conclusions might take into consideration the universality of the findings. I have several other comments and recommendations.
From the data, it seems that although there can be multiple entrances to a nest, functionally each nest has only two. Clusters of openings in close proximity (Figure 1: less 15 cm), or more separate (>15 cm). Within a cluster, it seems irrelevant which actual entrance is used (see Results, lines 206-214). Thus, at times I found it confusing when the authors mentioned entrance as to whether this meant separate clusters or not. For example, in the Methods on site fidelity, it was not clear until I got to Results as to what fidelity meant.
Figure 2: Only the red line appears on my copy.
The authors argue that multiple entrances increase food encounter probabilities (at a cost of reduced rapid recruitment to any single found one). It is a bit confusing again in the writing, but it seems that often colonies are just foraging from one entrance (cluster). How often did this happen? Was it the same pattern across all 5 colonies? It seems possible that on daily time scale, colonies switch back and forth between foraging strategies. Any idea why? Were some of the colonies more distant from competing neighbors, such that encounter probability might be more important than the ability to rapidly recruit many nestmates?
Three comments on the model.
- When food appears randomly in space, what would be the recruitment patterns across colonies that always foraged from one cluster, >1 cluster, or as in the field, a variable number of clusters (some trials one, other trials >1).
- It is not clear how in the model, ants ‘found’ the food. It would seem like where and how the foragers searched in space would critically affect the results (see Denton & Nonacs, 2018, Anim Behav 141:77–84).
- I found it very interesting that the authors suggested that their field results differed from previous lab results in that in the lab, foragers used the same trail up to a decision point. In contrast, in the field the foragers left from different places and no such decision point occurred (or at least was within the nest). This would seem easily testable in the model by adding some within-nest recruitment factor such that a returning forager increases the chances that recruited ants use its entrance.
Reviewer 2 Report
In this study, Lehue et al. perform a field study coupled to am agent-based model to examine the effect of nest entrance number on collective foraging in Myrmica rubra. They report that Myrmica rubra often show multiple nest entrances, usually clustered into one or two distinct groups. They further report that this influences collective foraging, with having multiple entrance clusters, and to a lesser extent more entrances, results in a more even distribution of foragers amongst feeders. They also report strong individual worker fidelity to food sources and nest entrances.
GENERAL REMARKS AND OVERVIEW
This study asks an important question, and is a much-needed fieldwork counterpoint to the lab studies performed by Lehue, and other researchers, on nest entrance number. Too few researchers (myself included) study the same question in the lab and in the field, and this effort is to be highly applauded. The manuscript is generally well written, with a few English issues which do not impede understanding. The introduction covers the relevant literature well, as does the discussion (see minor comments for a few additional suggestions). I also tend to believe the conclusions drawn from the fieldwork.
However, some aspects of this study are unsatisfying. The fieldwork suffers from very low sample-sizes in terms of the cluster number/entrance number data: only 5 colonies are studied. This leaves the work woefully underpowered. While some colonies are studied multiple times, this is not controlled for statistically, casting doubt on much of the field results. This removes a major aspect of the paper, leaving the empirical work an interesting natural history description (the individual worker fidelity data, and the nest entrance clumping data is solid). Whether this alone is appropriate for Insects I leave up to the editor.
I also have several issues with the agent-based model, including the assumptions made, its reporting, and its validation. More broadly, I do not see what major insights the ABM adds which could not be derived via a verbal argument (i.e. verbal model). I should say that I, myself, am also guilty of producing such models occasionally.
I know that extensive reviewer comments can be disheartening and frustrating. I hope the authors take my comments as evidence of my interest in their work. I consider this line of research productive and interesting, and wish to encourage it to be the best it can be.
FIELD WORK – DETAILED COMMENTS
- The sample size is really very small sometimes. 5 colonies, with 10 observations in total, is quite weak. Considering that one can manage an observation a day, it does not seem unreasonable to get 20 observations (i.e. 20 days of work, or one month including weekends). Doubling the number of studied colonies (from 5 to 10) would bring this into the realm of reasonable statistical analysis.
- Ln 105-106 – Time spent walking will strongly depend on how much experience the ant has with the food source. So when was this data taken? Or was this done on the marked ants, so we can be sure it’s the second trip for each of them? This is important since, as the authors mention in the discussion, experience will make the trips much faster. Similarly, in lns 113-116, we know that ants can form very long term memories and show months-long site fidelity to food sources (Rosengren and Fortelius 1986), and good memory after only one visit to a food source, which can last for more than a day (Piqueret Baptiste et al. 2019; Oberhauser et al. 2019). So is it reasonable to group observations here?
- Table S2 – how can the total number of marked ants observed entering a nest be higher than the total number of marked ants?
- Basic but important question – how did you ensure different nest entrances and different clusters were part of one colony?
- Figure 3 – this figure is overly complicated. The key information (Percent probability) is hidden by lots of other non-interesting stuff. Frankly, a stacked bar chart showing % ants returning to same hole/ different hole same cluster / different cluster would be better. Multiple bars could be given, one for each colony, if the author prefers.
STATISTICAL ANALYSIS – DETAILED COMMENTS
- All of the analyses are carried out without controlling for colony ID. This means that multiple readings have been taken from each colony, inflating the apparent sample size. This is not appropriate, as it results in underestimation of P values. This includes the results reported in lines 209, 210, 212, and 214. This is not simply statistical nit-picking: in this experiment, one oddly-behaving colony, sampled multiple times, may completely change the results of the study. I really do not see a good way around this issue: the sample size is simply too small, as I understand it. I admit to not being a very knowledgeable statistician, but this issue seems major to me, and will have to be very strongly tackled, or a clear explanation given for why this is not a major problem. The authors might be forced to collect more data, or to retract most of the conclusions made based on these data.
- Statistical analysis should not be carried out on model results (ln 258-260). This is because the sample size is arbitrary, and such statistical analysis is extremely strongly influenced by sample size. Rather, each permutation should be run at least 500 times, and then the effect size reported (visually and numerically).
AGENT-BASED MODEL – DETAILED COMMENTS
- Many critical parts of the model are not reported. This includes a clear table of all the variables chosen and their levels, the number of time steps for each model run, and the number of model runs per permutation of variable levels. I strongly recommend following the standardised ODD protocol for reporting agent-based models (Grimm et al. 2010).
- Model supplement:
lns 8-47 – much of this would be better in a table, with the levels chosen for each variable given.
lns 21-24 – how much pheromone is laid per step?
ln 26 – please refer to the figure here, to help the reader. By the way, the figure is really excellent! I rarely see such clear explanations for aspects of an ABM.
ln 29 – how was this value chosen? Why? Support with references to empirical work.
ln 36 – this is a key assumption of the model. However, is it reasonable to have one number of potential recruits per nest, not per entrance/cluster? If yes, please support. If no (and observations from within the lab nests from previous work by Lehue might help here), then this needs to be reconsidered).
ln 50 – so each time step is equivalent to one second? This should be stated. Is this reflected in the pheromone evaporation and distances travelled between feeders? How many time steps does a run last for?
ln 62 – this is also an important assumption of the model. Some studies suggest that memories and pheromones add a bias (von Thienen et al. 2016). Others suggest that, when memory is present, it overrides pheromone trail following (Grüter et al. 2011), or does so depending on other information available (Czaczkes et al. 2019). These decisions might be critical to the functioning of the model, so should A) be made explicit, and B) ideally be tested in a stability analysis (see below).
- The model lacks a sensitivity analysis. This is a systematic testing of assumed variables and other aspects of the model (world size, run length, procedure order), to ensure that the major conclusions drawn are robust to these (often arbitrary) decisions. Of course, every combination cannot be tested, but each assumed variable should be modified in isolation (on the background of a standard variable set) to ensure that the model doesn’t behave in a weird manner when some arbitrary assumptions are changed.
- A figure giving the decision-rules of each ant at each time step (as a flow diagram) would be useful. At line 149 this should be referred to, and (importantly) the reader directed to the detailed description in the supplement.
- I wonder whether the extension the authors chose (clumps of depleting food) is the most interesting one? What does it represent, biologically? Maybe seeds. A different extension, which would be (to me) more interesting would be limited-rate feeders, simulating aphid patches, which deplete and refill (at the same or different rates), as done in (Czaczkes et al. 2015). Another interesting extension would be a changing environment – again looked at often, for example in (Czaczkes et al. 2015) – how do multiple nest entrances help or hinder ants to keep track of changes in dynamic environments? Yet another would be randomly distributed single-shot food sources (simulating prey or dead insects). These all would provide interesting ecological insights. This need not be done here: it is just a suggestion.
- Ln 248-250 – isn’t this a trivial result? More entrances > more ‘closest’ food sources. If each entrance ‘selects’ its closest foods source, more food sources will be selected by more entrances, and more strongly by more clusters (which are by definition further away).
MINOR COMMENTS
- Title – personally, I prefer a more results-oriented title. I would rather be told what was found, not what was done. For example “Ant colonies with more nest entrances distribute their foraging more evenly” or similar.
- Ln 13 – ‘related collective structures’ unclear.
- Ln 14 – specify Myrmica rubra – “red ant” refers to different species, depending on location.
- Ln 14 – ‘situated’ replace with ‘located’
- Ln 22 – replace ‘appeared as’ with “seem to be”. I will stop making minor English corrections at this point, but would be willing to help with this for the next version of this manuscript.
- Ln 75 – here a quick reference to the benefits of polydomy, and one reference, would be good. Such info is found in the discussion, but would be helpful here.
- Ln 102 – 0.3M, really, so low? Out of interest, why choose such a low molarity? In the field, I find many species reject such low molarities.
- Ln 192-194 – I think this paper is relevant here and elsewhere, and will be interesting to the authors: (Latty and Beekman 2013). It is a rare field test of collective decision-making and symmetry breaking in a wide variety of ant species.
- Figure S2 is not really needed, as far as I can tell.
- Figure 2A – only the red line is present, the rest have disappeared.
- Figure 2B – if there are only 5 colonies, how can there be more than 5 dots? Answer: pseudoreplication (see main comment)
- Ln 226 – or were cleaned while in the nest
- Lns 290-291 – this assumes a single pool of recruits per nest. Is this a reasonable assumption?
- Ln 315-316 – this is an interesting result.
- Ln 319-20 – this is interesting. Where was the data for this?
- Ln 323-324 – this is probably important information for justifying some model assumptions.
- Ln 344-348 – this was a lab study, which probably poorly reflects behaviour in the field. I don’t think it is reasonable to compare these two studies, and claim that this is a species difference. See (Latty and Beekman 2013) for a more reasonable comparison to multiple other species, or compare Lehues own good work in the lab to contrast lab vs field.
- Ln 364-5 – not necessarily: position image matching could also do this, without a mental (not mind) map.
- Ln 369-70 – see (Czaczkes et al. 2015) for a model showing this explicitly in an ant example, and (Schürch and Grüter 2014) for a model showing this in honeybees.
- Ln 370-372 – shown experimentally in (Czaczkes et al. 2016)
- Ln 379 – or more willing to give up a food source when other information is available, eg (Czaczkes et al. 2019). This need not be cited! Just mentioning this in case the authors are interested.
REFERENCES CITED
Note:I realise I have self-cited quite strongly. Mentioning a paper here does not imply a demand for citation in the manuscript – I simply know my own work best.
Czaczkes TJ, Beckwith JJ, Horsch A-L, Hartig F (2019) The multi-dimensional nature of information drives prioritization of private over social information in ants. Proc R Soc B 20191136:. http://dx.doi.org/10.1098/rspb.2019.1136
Czaczkes TJ, Czaczkes B, Iglhaut C, Heinze J (2015) Composite collective decision-making. Proceedings of the Royal Society B: Biological Sciences 282:
Czaczkes TJ, Salmane AK, Klampfleuthner FAM, Heinze J (2016) Private information alone can cause trapping of ant colonies in local feeding optima. J Exp Biol 219:744–751. https://doi.org/10.1242/jeb.131847
Grimm V, Berger U, DeAngelis DL, et al (2010) The ODD protocol: A review and first update. Ecological Modelling 221:2760–2768. https://doi.org/10.1016/j.ecolmodel.2010.08.019
Grüter C, Czaczkes TJ, Ratnieks FLW (2011) Decision making in ant foragers (Lasius niger) facing conflicting private and social information. Behav Ecol Sociobiol 64:141–148. https://doi.org/10.1007/s00265-010-1020-2
Latty T, Beekman M (2013) Keeping track of changes: the performance of ant colonies in dynamic environments. Anim Behav 85:637–643. https://doi.org/10.1016/j.anbehav.2012.12.027
Oberhauser FB, Schlemm A, Wendt S, Czaczkes TJ (2019) Private information conflict: Lasius niger ants prefer olfactory cues to route memory. Anim Cogn 22:355–364. https://doi.org/10.1007/s10071-019-01248-3
Piqueret Baptiste, Sandoz Jean-Christophe, d’Ettorre Patrizia (2019) Ants learn fast and do not forget: associative olfactory learning, memory and extinction in Formica fusca. Royal Society Open Science 6:190778. https://doi.org/10.1098/rsos.190778
Rosengren R, Fortelius W (1986) Ortstreue in foraging ants of the Formica rufa group — Hierarchy of orienting cues and long-term memory. Insect Soc 33:306–337. https://doi.org/10.1007/BF02224248
Schürch R, Grüter C (2014) Dancing bees improve colony foraging success as long-term benefits outweigh short-term costs. PLoS ONE 9:e104660. https://doi.org/10.1371/journal.pone.0104660
von Thienen W, Metzler D, Witte V (2016) How memory and motivation modulate the responses to trail pheromones in three ant species. Behav Ecol Sociobiol 70:393–407. https://doi.org/10.1007/s00265-016-2059-5
Round 2
Reviewer 2 Report
I have responded where needed to the author responses. My comments are highlighted in yellow.
